# Effects of a Structured Multicomponent Physical Exercise Intervention on Quality of Life and Biopsychosocial Health among Chilean Older Adults from the Community with Controlled Multimorbidity: A Pre–Post Design

**DOI:** 10.3390/ijerph192315842

**Published:** 2022-11-28

**Authors:** Rafael Pizarro-Mena, Samuel Duran-Aguero, Solange Parra-Soto, Francisco Vargas-Silva, Sebastian Bello-Lepe, Mauricio Fuentes-Alburquenque

**Affiliations:** 1Facultad de Odontología y Ciencias de la Rehabilitación, Universidad San Sebastián, Sede Los Leones, Santiago 7500000, Chile; 2Facultad de Ciencias para el Cuidado de la Salud, Universidad San Sebastián, Sede Los Leones, Santiago 7500000, Chile; 3Departamento de Nutrición y Salud Pública, Universidad del Bío-Bío, Chillan 3780000, Chile; 4Escuela de Fonoaudiología, Universidad de Valparaíso, Viña del Mar 2520000, Chile; 5Centro de Investigación del Desarrollo en Cognición y Lenguaje (CIDCL), Universidad de Valparaíso, Viña del Mar 2520000, Chile; 6Escuela de Salud Pública, Facultad de Medicina, Universidad de Chile, Independencia 8380000, Chile

**Keywords:** physical activity, physical exercise, older adults, biopsychosocial effects, quality of life

## Abstract

Structured multicomponent physical exercise (PE) for older adults, with a combination of strength, aerobic, flexibility, and balance exercises, has been shown to have benefits for physical, cognitive, social, and metabolic functioning, as well as counteracting chronic pathologies and geriatric syndromes. However, little is known about the effect of these interventions in Chilean older adults. Our objective was to determine the effect of a structured multicomponent PE intervention on the quality of life (QoL) and biopsychosocial factors of community-living older adults. We conducted a pre–post intervention without control group, with a face-to-face structured multicomponent PE intervention (cardiovascular, strength/power, flexibility, static and dynamic balance, other psychomotor components, and education), based on FITT-VP principles (frequency, intensity, type, time, volume, and progression of exercise), at moderate intensity, 60 min per session, three times per week, and 12 weeks in duration, among 45 persons with an average age of 70.74 years. Participants were evaluated at the beginning and end of the intervention with different instruments of comprehensive gerontological assessment (CGA). Post intervention, participants (83.70% average attendance) significantly improved scores in QoL, biological and biopsychosocial frailty, sarcopenia, functionality in basic, instrumental, and advanced activities of daily living, dynamic balance, cognitive status and mood, systolic and diastolic blood pressure, weight, body mass index, strength and flexibility clinical tests of lower and upper extremity, aerobic capacity, agility, and tandem balance. The indication and prescription of structured multicomponent PE based on FITT-VP principles, as evaluated with the CGA, improved the QoL and biopsychosocial health of older adults. This intervention could serve as a pilot for RCTs or to improve PE programs or services for older adults under the auspices of existing public policy.

## 1. Introduction

Older adults are a sector of the population that is growing rapidly worldwide, with a projected population of 1400 million in the year 2030, representing an increase of 65% in 15 years [1].

In this age group, the normal aging process (with morphological and physiological changes in different organs and systems) [2], as well as the biopsychosocial modifications and changes in the physical–functional condition, generate a series of intervention needs translating into a decrease in the activities of daily life and physical, functional, cognitive, and social functioning and performance, as well as into physical inactivity and a sedentary lifestyle, chronic pathologies, geriatric syndromes, and disability [3].

Physical inactivity, which reaches 16.5% in the European population aged 55 and over [4], is generally understood as the spectrum of any decrease in body movement that produces a reduction in energy expenditure toward the basal level (including aging). Specifically, physical inactivity in people aged 18 and over, such as noncompliance with the minimum international recommendations for physical activity (PA) for the health of the population [5], has been identified as a cause or risk factor for around 35 chronic pathologies and geriatric syndromes, including most of the 10 main causes of death in the North American population [5]. In addition, older adults are among the most sedentary groups, understood as a lack of movement during waking hours throughout the day [6], reaching 67% of the older population who present sedentary behavior for more than 8.5 h a day, which is associated with poor health outcomes and all-cause mortality regardless of PA levels [7].

All these changes and effects with normal and pathological aging become important opportunities for intervention, with the regular practice of PA and physical exercise (PE) being one of the most recommended interventions at present.

As concepts, PA and PE are used interchangeably, but they are not equivalent, since PA is defined as any body movement produced by skeletal muscles that requires energy expenditure. On the other hand, PE is a variety of PA, which is planned, structured, repetitive, and performed with or without the explicit intention of maintaining or improving one or more components of the physical–functional condition (e.g., aerobic capacity, strength and muscle power, balance, coordination, and flexibility) [3].

Accordingly, multicomponent (or multimodal) PE interventions have been defined as a combination of strength/power, aerobic, flexibility, and balance exercise, to promote health [8]. Multicomponent PE interventions should be differentiated from multicomponent interventions (complex, multimodal, multidomain, or multidimensional) that correspond to a series of nonpharmacological, behavioral, and environmental strategies [9] that combine multiple health promotion activities, including cognitive stimulation, PA and PE, education, risk factor management, nutrition and caloric restriction, psychological wellness and stress management, leisure activities, and social support [10]. However, they are complementary, since multicomponent interventions can also incorporate multicomponent PA or PE, with kinesiologists/physiotherapists having the greatest competencies in this approach among professionals of an interdisciplinary gerontological intervention.

In recent years, multicomponent PE interventions have shown the greatest benefits in improving the physical–functional condition including balance/agility, strength of upper (UE) and lower (LE) extremities, and cardiorespiratory capacity [11] for physical, functional, cognitive, social, and metabolic function and performance [3,8,12]. In addition to the benefits mentioned, PE interventions have been shown to counteract chronic pathologies and geriatric syndromes [3].

For instance, in a recent scoping review on PE programs designed for older adults (87 RCTs), it was identified that structured and multicomponent PE programs have a strong positive impact on falls, intrinsic capacity, and physical functioning [12]. Additional positive impacts were identified for various chronic pathologies (osteoporosis, osteoarthritis, diabetes, obesity, hypertension, cardiovascular disease, cancer, and Parkinson’s) and geriatric syndromes (sarcopenia, frailty, falls, immobility, depression, cognitive impairment, and dementia) [3].

Likewise, the reduction in or control of chronic pathologies through PE favors the reduction in and/or deprescription of drugs, since polypharmacy is a risk factor or cause of several geriatric syndromes [13], generating a vicious cycle. Consequently, both PA in general and multicomponent and structured PE play an important role as promotion and prevention strategies, which allows reducing costs in public health.

As far as the intervention itself is concerned, a recent systematic review of randomized PE clinical trials conducted among Latin American populations [14] found that interventions have focused mainly on cardiovascular exercise and strength, leaving aside other components of the physical–functional condition. Sessions had a duration of 30–60 min, frequency of 2–3 times a week, and intervention duration between 2 and 6 months, they were led by sports science professionals with predominance in the community, and less than half included a warm-up and/or cool-down stage in each session. Of the 101 interventions analyzed, only five were conducted in the Chilean population, none of them performed a multicomponent PE intervention, and almost the same number of investigations reported improvements in psychosocial factors and quality of life (QoL).

In this context, and in Chilean public policy, there are two government programs that carry out multicomponent PE in older adults. The first, a 3 month intervention on self-reliant older adults in the community, was carried out in primary healthcare centers, under the auspices of the Ministry of Health [15]. The second, a 6 month intervention on self-reliant and mildly dependent older adults in the community, was carried out in community day centers for older adults, under the umbrella of the National Service for the Older Adults [16]. In both, the PE was performed by a kinesiologist/physiotherapist. However, both programs have received very little research attention, and they lack a structured and protocolized intervention of multicomponent PE following the principles of the FITT-VP (frequency, intensity, type, time, volume and progression of exercise) [17]. This is an opportunity in the design, implementation, and evaluation of structured multicomponent PE intervention strategies with a gerontological emphasis, which can contribute to both research and the improvement of programs and services under the umbrella of existing public policy. Furthermore, they can incorporate other components such as psychomotor in addition to the traditional components recommended by international recommendations [3], given that, as people grow older, other aspects of psychomotricity should be incorporated into interventions, thus contributing to active, healthy, and satisfactory aging [18].

Regarding the evaluation of structured multicomponent PE interventions, the effects of PE are traditionally evaluated by means of scales, questionnaires, or clinical tests of the physical–functional condition, leaving aside the biopsychosocial comprehensiveness facilitated by the comprehensive gerontological assessment (CGA). The CGA is the cornerstone of interdisciplinary evaluation in gerontology, which is made up of scales, indices, questionnaires, surveys, and clinical and performance tests, evaluating multiple domains of the biomedical, physical–functional, cognitive, psychic, social, environmental, and QoL spheres of the older adults. The CGA is a set of simple, low-cost, and replicable evaluation tools used in different contexts, levels of care, programs, and services involving older adults [19].

Likewise, in the field of gerontology as a science of aging, the intervention must consider the integrality of the older adults, encompassing various spheres, while favoring more complex and comprehensive interventions. Accordingly, QoL has quickly become a standard for measuring the results of gerontological and long-term care services, representing one of the most important objectives in any intervention with older adults [20]. In a structured multicomponent PE intervention, QoL can be understood as an individual’s self-perception of their position in life, in the context of the culture and value systems in which they live, and in relation to their goals, expectations, norms, and concerns, considers multidimensional aspects of physical health, psychological state, level of autonomy, social relations, beliefs, and the relationship with the outstanding characteristics of the environment [21].

The objective of this study was to determine the effect of a structured multicomponent PE intervention on QoL and biopsychosocial factors of community-dwelling older adults.

## 2. Materials and Methods

### 2.1. Study Design

We conducted a pre–post intervention design without control group [22].

### 2.2. Sample

The sample comprised 52 people ≥60 years of age, with treated and compensated multimorbidity, from a community center in an urban city in central Chile, who met often for social activities (but did not participate in PE), who were having their regular health checkups in primary healthcare, and who were invited to participate in the research. Inclusion criteria were as follows: (a) 60 years or older, for both sexes, (b) self-reliant, (c) able to see and hear well enough to participate in the intervention, (d) without major physical and/or cognitive disease/disability that would affect participation, and (e) having completed the intervention with a minimum of 60% attendance. Participants who were performing a similar intervention in the same follow-up period were excluded. Participants were evaluated using the CGA battery, which includes evaluation instruments that are used regularly in healthy older adults, with comorbidities and/or geriatric, self-reliant, or mildly dependent syndromes, from the community or primary healthcare setting, as a regular part of intervention in gerontology and existing local public policy [23] (described in detail below). They intervened for 12 weeks with the PE intervention, and then reassessed using the same CGA battery. During follow-up, seven people withdrew from the study; thus, 45 participants completed the intervention/assessment (83.70% average attendance; see Figure 1). The sample size was calculated using GPower (10) for pre–post statistical analysis (Wilcoxon or paired-samples *t*-test); the parameters included a 0.8 effect size, with 0.05 α error probability and 0.95 β error probability. The research was approved by the ethics committee of the Eastern Metropolitan Health Service (Servicio de Salud Metropolitano Oriente); all participants signed an informed consent form.

### 2.3. Assessment and Variables

The evaluation of the intervention was carried out in three sessions at the beginning and three sessions at the end of the intervention by a separate team of health professionals trained in the application of the CGA. The sociosanitary characterization form and all CGA instruments were administered using Google Forms, which allowed timely tabulation and validation, while minimizing errors in the transfer of information from paper to computer. For the application of the scales, questionnaires, surveys, and indices, the professionals personally consulted each older adult. For the evaluation of the clinical tests, the following tools were implemented: chair without an armrest, table, pencil, masking tape, scale, stadiometer, dumbbell, metal tape measure, stopwatch, cone, dynamometer, pressure gauge, and a saturation meter.

The variables and evaluation instruments of the CGA battery are presented below [19].

*Quality of life (QoL)* was the primary study variable and was evaluated using the EuroQol visual analogue scale (EQ-VAS) questionnaire validated in the Chilean population; higher scores indicated a higher QoL, with a maximum score of 100 [24].

*Frailty.* Biological frailty was evaluated with the FRAIL scale validated in the Mexican population, scored as follows: without frailty, 0 points; pre-frailty, 1–2 points; fragile, 3–5 points [25]. Biopsychosocial frailty was evaluated using the Tilburg Index validated in the Spanish population, scored as follows: no frailty, 0–4 points; fragile, 5–15 points [26].

*Sarcopenia* was evaluated using the SARC-F scale validated in the Mexican population, scored as follows: without sarcopenia, 0–3 points; sarcopenia, 4–10 points [27]. In addition, hand dynamometry (baseline digital model 12-0286) was used to assess force (handshake) in kilograms, with the following cutoff points for sarcopenia in the Chilean population: <27 kg (men) and <15 kg (women) [28].

*Functionality.* Basic activities of daily living (ADL) were evaluated using the Barthel Index, with a higher score indicating greater independence in ADL (maximum score of 100) [29]. Basic, instrumental, and advanced ADLs were evaluated using the Technology—Activities of Daily Living Questionnaire (T-ADLQ) questionnaire validated in the Chilean population, with a lower score indicating greater independence in ADLs [30].

*Balance.* Static balance and risk of falls were evaluated using the one-leg balance (OLB) clinical test; three attempts were made with the best of the three considered and categorized as follows: normal, ≥5 s; altered, <4 s [31]. Dynamic balance and risk of falls were evaluated using the timed up and Go (TUG) clinical test, categorized as follows: normal, ≤10 s; slight risk of falling, 11–20 s; high risk, >20 s [32]. In addition, the cognitive (TUGc) and manual (TUGm) variants were added, with shorter time indicating greater dynamic balance [33].

*Physical–functional condition* was evaluated using the senior fitness test (SFT), a clinical test battery that incorporates the following tests: chair stand test, sit and stand in 30 s (LE strength), arm curl test, push-ups in 30 s (UE strength), 2 min step test (aerobic capacity), chair sit and reach test (in which two attempts were allowed and the best was attempt was noted; LE flexibility), back scratch test (in which two attempts were allowed and the best was attempt was noted; UE flexibility), and 8 foot up and go test (in which two attempts were allowed and the best was attempt was noted; agility) [34]. For the first five tests, a higher value indicates better performance; for the last test, a lower value indicates higher performance. Furthermore, the short physical performance battery (SPPB) clinical test was applied, which incorporates three clinical tests: balance in three positions (side-by-side stand, semi-tandem stand, and full tandem stand), walking speed in 4 m (two attempts were allowed and the best was attempt was noted), and sitting and standing five times, with a higher score indicating higher physical performance (maximum of 12 points) [35].

*Cognitive status and higher functions* were evaluated using the short portable mental status questionnaire (SPMSQ) validated in the Spanish population, scored as follows: 0–2 errors, intact intellectual functions; 3–4 errors, mild cognitive impairment; 5–7 errors, moderate cognitive impairment; 8–10 errors, severe cognitive impairment [36]. In addition, the Montreal Cognitive Assessment (MoCA) validated in the Chilean population was applied, with a higher score (maximum score of 30) indicating higher cognitive performance [37]. Furthermore, the trail making test, parts A and B (TMT-A/TMT-B), which clinically tests attention and psychomotor speed and has been validated in the Argentine population, was added, with lower time indicating higher neuropsychological functioning [38].

*Mood* was evaluated using the Yesavage geriatric depression scale (GDS), which has been validated in the Spanish population, scored as follows: normal score, 0–5 points; mild depression, 6–9 points; established depression, 10–15 points [39].

*Sleep* was evaluated using the insomnia severity index (ISI) validated in the Spanish population, scored as follows: absence of clinical insomnia, 0–7 points; subclinical insomnia, 8–14 points; moderate clinical insomnia, 15–21 points; severe clinical insomnia, 22–28 points [40]. The Epworth Daytime Sleepiness Questionnaire (ESS), which has been validated in the Colombian population, was also applied, scored as follows: normal, 0–9 points; marginal sleepiness, 10–12 points; excessive sleepiness, 13–24 points [41].

*Food quality* was evaluated using the food quality survey for the older adults (ECAAM, acronym in Spanish) validated in the Chilean population, with a higher score indicating higher food quality [42].

*Health empowerment* was evaluated using the health empowerment scale for older adults (HES), validated in the Argentine population, with a higher score indicating greater health empowerment [43].

*Sociodemographic, anthropometric, and clinical variables.* Sociodemographic variables (age, sex, schooling, marital status, use of technical aids (cane), number of pathologies, number of drugs, laterality, and number of falls in the last year) were evaluated. In addition, anthropometric variables (weight, height, body mass index (BMI) [44], waist and [45], neck circumference [46]) and vital signs (systolic and diastolic blood pressure [47], heart rate, and saturation) were measured.

### 2.4. Intervention

The structured intervention lasted 12 weeks, three sessions per week (Monday, Wednesday, and Friday), for a total of 36 sessions. Each session was structured as follows: 10–15 min warm-up; 40–45 min multicomponent intervention including cardiovascular exercise, strength/power, flexibility, and static and dynamic balance exercises [8], prescribed according to FITT-VP principles [17], as well as other psychomotor components, together with education on PE and healthy lifestyles; 5–10 min cooldown. We followed previous recommendations for PE for older adults [48,49], and three progressive cycles based on these principles were performed. Specifically, the intensity of cardiovascular and strength exercise was progressively managed at a moderate intensity in line with the literature [50,51,52].

The face-to-face intervention and in-group activities were carried out between March 2019 and January 2020 by a kinesiologist/physiotherapist trained in PE topics for older adults including good treatment, interpersonal relationships between peers, and person-centered care, with a master’s degree in gerontology and geriatrics. This intervention was carried out in a community center; the space was wide, roofed, and flat.

Different instruments were used for the intervention: chairs without armrests, tables, dumbbells and anklets of different weights (kilograms), elastic bands and tubes with different resistance, wooden sticks, yoga belts (flexibility), hoops, cones, balls, and balloons.

The intervention protocol was written and systematized prior to the intervention; during the intervention weeks, the main researcher and the professional in charge regularly contrasted what was planned and executed, to avoid deviations. The protocol is summarized and outlined in Figure 2.

### 2.5. Statistical Analysis

Pre- and post-intervention descriptive analyses are presented as the mean and standard deviation for continuous variables, and as the frequency and percentage for categorical variables. To determine differences between the pre- and post-intervention evaluations, the Wilcoxon or paired-samples *t*-test was calculated, according to distribution, and the change (delta) and its standard deviation were measured. Cohen’s d (d) was used to measure effect size, with values >0.8 interpreted as a high magnitude of the effect. The Shapiro–Wilk test was performed to evaluate whether variables had a normal distribution with a value of *p* > 0.05. All analyses were performed with the statistical software STATA MP version 17.3.

## 3. Results

Participants were on average 70.74 years of age; the majority were women, university students, and married. Almost all (93.3%) did not need technical aids; on average, they had five pathologies and used three drugs (Table 1).

The pre–post intervention comparison is presented in Table 2. A significant increase was observed in the scores for the QoL (Δ = 7.8, d = 0.45), basic ADLs (Δ = 1.77, d = 0.43), and cognitive status (Δ = 1.45, d = 0.32). Moreover, biological fragility (Δ = −0.36, d = −0.45), biopsychosocial frailty (Δ = −1.13, d = −0.45), sarcopenia (Δ = −0.64, d = −0.41), basic, instrumental, and advanced ADLs (Δ = −2.06, d = −0.20) and mood (Δ = −0.98, d = −0.61) also showed significant improvements, i.e., decreases in scores (Table 2).

With respect to clinical aspects, post intervention, seated systolic blood pressure decreased significantly (Δ = −11.73, d = −0.65), as did seated diastolic blood pressure (Δ = −6.56, d = −0.57), standing systolic arterial (Δ = −8.24, d = −0.44), and dynamic balance with cognitive task (Δ = −2.07, d = −0.44). Weight and BMI also decreased significantly in the intervention group (Δ = −0.91, d = −0.08; Δ = −0.49, d = −0.12, respectively) (Table 3).

Lastly, in relation to the physical–functional condition evaluations, post intervention, differences were observed in almost all the tests carried out. Clinical test scores significantly improved in LE strength (Δ = 1.49, d = 0.40), UE strength (Δ = 1.58, d = 0.38), aerobic capacity (Δ = 11.22, d = 0.55), LE flexibility (Δ = 1.24, d = 0.14), UE flexibility (Δ = 2.18, d = 0.20) and agility (Δ = −0.90, d = −0.45) for the SFT, as well as in clinical full tandem stand (Δ = 1.04, d = 0.36), sitting and standing five times (Δ = −3.29, d = −0.81), and performance classification of the SPPB (Δ = 1.11, d = 0.63) (Table 4).

## 4. Discussion

The objective of this study was to determine the effect of a structured multicomponent PE intervention on QoL and biopsychosocial factors of community-dwelling older adults, using a pre–post intervention design without control; the main result showed that a structured multicomponent PE intervention conducted on a sample of Chilean older adults from the community, with controlled and compensated multimorbidity, improved their QoL, biopsychosocial health, and clinical and physical–functional condition.

In our sample, the older adults presented multimorbidity (two or more chronic pathologies), with an average of four pathologies that were being treated and compensated at the time of the intervention, since they were up to date with their regular care checkups in a primary healthcare setting, which is similar to what was reported in the Chilean older adult population according to the last National Health Survey of Chile (2016–2017) [53], where 74% of women and 54% of men had multimorbidity. Regarding the drugs consumed by the older adults in the intervention, it was an average of three. According to the survey cited above, nine out of 10 older adults consumed at least one drug. In addition, polypharmacy (five or more drugs) was recorded in 37% of older adults in Chile, with a predominance of women and participants with less schooling [13]. For every 10 older adults in our group, three were men and seven women, which is similar to what professionals regularly see in the regular practice of PA and PE with older adults from primary healthcare settings and socio-community programs in Chile, where the participation of men ranges between 1 and 3 for every 10 older adults, whereas the participation of women in PE programs in Latin America was previously identified as 79% [14].

Our main variable was QoL, for which we observed significant improvements. According to the literature, to improve QoL for older adults using PE, muscular strength exercises (at moderate intensity) and/or aerobic PE (at moderate intensity) should be prescribed according to FITT principles, in 60 min sessions, three times a week, for at least 3 months duration, without greater clarity in relation to volume and progression (VP) [19]. These suggestions correspond to several of the parameters addressed in our intervention, which did include VP. Using the principles of FITT-VP not only guarantees the specificity of dose similar to a pharmacological intervention [54], but also enables medical deprescription, e.g., for depression due to a greater effect of PE compared to available drugs [3], in addition to facilitating the planning of time, necessary to improve health.

In relation to our main results, it has been described that structured and multicomponent PE programs have a strong positive impact on falls, intrinsic capacity, and physical functioning (physical and cognitive/emotional domain, as well as in the social domain) [12], as well as on various chronic pathologies and geriatric syndromes [3], which supports our findings in relation to significant improvements in blood pressure, weight, BMI, sarcopenia, biological and biopsychosocial frailty, functionality (basic, instrumental, and advanced ADLs), cognitive status, and mood.

We found no significant improvements in attention and psychomotor speed, sleep (insomnia and daytime sleepiness), eating quality, and health empowerment. In relation to sleep, our prescription parameters were within previously described parameters to achieve effects on sleep in the older adult population of the community [55]. In relation to the quality of food, our intervention only contemplated some educational activities in this regard, and it is possible that education in healthy eating should be reinforced in future interventions and/or complemented with an intervention by the professional nutritionist. In relation to health empowerment, the participants at the beginning had a high level of empowerment (close to the maximum score of the evaluation instrument), which did not vary greatly with the reassessment, possibly because they were regular users of primary healthcare where there are regular promotion strategies and/or they were very aware from the beginning of the intervention of the benefit of PE. However, it may be necessary to increase the intervention time and/or adjust the prescription parameters according to the FITT-VP principles to achieve effects on these variables, as well as introduce actions for its improvement; therefore, it may be interesting to continue investigations in the future.

Among the most used multiple components for PE among older adults are strength and cardiovascular exercise [14]; however, in our intervention, we incorporated all components of a structured multicomponent exercise (cardiovascular, strength/power, flexibility, and static and dynamic balance) [8], in addition to a structure for each session (warm-up, central, and cool-down stages), which has been identified to a lesser extent in interventions in Latin America [14]. Our intervention referenced other similar interventions in terms of design, recommendations, prescription, and types of exercises, as supported by scientific evidence [50,51,52,56]. As a novel aspect, our intervention provides, in addition to the traditional sliding and walking, different exercises used in a sitting position for cardiovascular exercise: using the mobility of the upper and lower extremities at the same time, symmetrically or asymmetrically, associated with rhythm, coordination, and breathing; adding the use of a cane (simulate rowing); exercising with static gait in a seated position. In addition, we incorporated psychomotor components into the intervention: eye–hand and eye–foot coordination, laterality, body scheme, and double tasks (the main functions incorporated in the double tasks were memory, attention, concentration, language, orientation, and executive functions). Overall, this contributed to a more comprehensive structured multicomponent PE intervention in the older adults.

Consequently, the incorporation of more components (flexibility and static/dynamic balance), as well as gerontopsychomotor rehabilitation, should be contemplated in future studies that include interventions for persons of older ages and/or with decreased cognitive status, since they could generate effects in more spheres of wellbeing.

In the current study, the greatest magnitude of effects was observed in mood measured with GDS, sitting blood pressure, LE strength and physical performance measured with SPPB, and cardiorespiratory aerobic capacity measured with SFT. In a recent meta-analysis that included 97 RCTs, it was identified that PE for older adults produces a moderate improvement in depression and depressive symptoms [57]. In addition, aerobic PA has been reported to increase gray-matter volume ratio and improve white-matter spatial structure in the brain, leading to greater functional connectivity in brain regions associated with major depression [58]. Thus, it is relevant to continue investigating the effects of multicomponent PE, including aerobic exercise, for the improvement of depression among older adults.

Although significant changes in almost all and effect sizes in some clinical tests from the SPPB and SFT batteries were observed in our research, the SFT for older adults living in the community physical conditioning may be the most appropriate clinical battery, since it evaluates all the components addressed by the multicomponent PE (e.g., cardiovascular, balance/agility, strength and flexibility of UE and LE). It has also been described that SPPB is less sensitive to changes in physical performance over time and is better suited to biologically fragile older adults [59], although this was not the reality of our participants.

Regarding the self-report evaluation instruments used in our research to evaluate the CV variables, biological and biopsychosocial frailty, sarcopenia, functionality, mood, sleep, quality of food, and health empowerment, and in order to reduce the possibility of bias, instruments regularly used in the practice of gerontology and geriatrics were applied. When possible, the versions validated in the Chilean and/or Hispanic populations were used, and their application protocols were followed. All our evaluation instruments were established on the Google Forms platform, which minimized the possibility of error. In addition, each evaluation instrument was tailored by the professional to each older adult, who was also trained in the application of all evaluation instruments of the CGA.

### 4.1. Implications for Future Practice

The recently published international recommendations for PE in the older adults by the International Exercise Recommendations in Older Adults (ICFSR) [3], should be a mandatory reference document for all professionals linked to the sciences of PA, i.e., those who work with healthy older adults, as well as those with chronic pathologies and geriatric syndromes, to promote health promotion and prevention or as a therapeutic agent, since they establish scientific evidence and a common language in terms of these interventions. In addition, they are the basis for the design and implementation of structured multicomponent PE intervention in older adults, in socio-community or clinical contexts, at different levels of healthcare (primary, secondary or tertiary), both in research and in public policy in different programs that are implemented for older adults. At the same time, they allow monitoring the state of health and functioning of older adults.

Recommendations provided by the WHO should be used as a minimum recommendation to suggest multicomponent PE in older adults [49], whereas more specific prescriptions recently delivered should be used as a ceiling [3], e.g., increasing volume (frequency, intensity, and exercise time), progression, and complexity [48].

In the context of Chilean public policy, our protocolized and structured intervention can serve as a good alternative to improve other multicomponent physical exercise group intervention programs in older people, who have a similar intervention time of 3 months, as in the case of the More Self-Reliant Older Adults program, implemented by the kinesiologists of the primary healthcare centers in the community where older adults meet, across more than 300 cities of the country [15].

Within gerontology, the evaluation of structured PE should go beyond simply evaluating the physical–functional condition, as is traditional, instead using scales, questionnaires, and clinical tests such as the CGA as a cornerstone of the evaluation process and interdisciplinary work. The CGA evaluates multiple domains in an integral way (biomedical, physical–function, cognitive, emotional, social, environmental, and QoL), which are adjusted to the reality of older adults, allowing greater accessibility, enabling coordination between services, and reducing health costs [19].

Older adults should be encouraged to participate in multicomponent PE programs to support healthy, active, and optimal aging [49], to improve not only physical–functional condition, but also biopsychosocial health as a whole. Adherence can also be encouraged using messaging services (e.g., WhatsApp) or direct phone calls. This may be particularly beneficial for older adults in worse health or who are disabled [60].

Consequently, interventions should also been deepened through person-centered care, considering the interests and expectations of older adults [61] when performing PE. For example, orientating behavioral change through the application of the transtheoretical model, which recognizes stages of change, can motivate and generate actions at each stage, which favors health empowerment [62]. Professionals may also consider using an ecological model perspective for working with older adults, which considers internal and external factors, from the micro to the macro environment, to avoid barriers and favor facilitators [63]. This approach may facilitate greater biopsychosocial effects of PE, with kinesiologists/physiotherapists being the most ideal health professional to evaluate the effects, make suggestions, and intervene with PE, among healthy or sick older adults, as part of an interdisciplinary gerontology team.

### 4.2. Implications for Future Research

Future research on the effect of multicomponent EF on older adults should focus on achieving greater specificity when prescribing PE in groups of older people. Despite the fact that one of the qualities of the exercise prescription is to be individualized, the value of group activities in older adults cannot be underestimated, as was the emphasis of our intervention, having a direct impact on aspects such as participation, the social role, the significant activities, the sense of belonging, and consequently, the QoL. Therefore, specificity was sacrificed for an impact on a greater volume at the population level, a necessary criterion that points to the fulfillment of goals in terms of national aging.

Likewise, and from a broader perspective, these interventions could incorporate biochemical profile analysis and explore other aspects of the social sphere of the CGA, e.g., the potential effect on family and social networks, community participation, or perceived social support. Another interesting aspect of the CGA to explore includes spirituality, which was not addressed in the current study and where there is almost no evidence.

Furthermore, more specific analyses should be carried out in the biomedical, physical-functional, mental, and/or social spheres, as well as more complex analyses articulating all variables, which could reveal other effects not carried identified in the present analysis. New research can be complemented by a qualitative approach to triangulate quantitative and qualitative (mixed) studies and gain an in-depth understanding of the complexity and diversity of effects of structured PE among older adults living in the community.

Considering the context of the recent COVID-19 pandemic, interventions provided in synchronous, asynchronous, or hybrid/telehealth modalities can also be designed, implemented, and evaluated.

### 4.3. Strengths and Limitations

Our study had several limitations. The intervention time of 3 months did not allow long-term effects to be identified; however, this time was enough to show improvements in the results [3,14]. We also lacked some CGA assessment instruments that were validated for use among the Chilean population. That said, data collection tools were complemented with other instruments validated in other Spanish-speaking populations and considered in the public policy of these countries [23,64]. Selection bias cannot be discredited. Moreover, given the number of participants, an analysis differentiated by sex was not performed, as the sample constituted a higher proportion of women than men as typical in the local environment [14]; thus, it is also possible that our results were overestimated. Future studies should consider including a more rigorous sampling method, a larger intervention group, and the enrollment of a control group. These limitations restricted the ability of this study to show robust intervention effects, thus limiting generalizability.

We can also highlight several strengths of this research. First, the intervention protocol designed and applied, as a pilot, may be useful for future RCTs or the improvement/implementation of programs and public policies on the subject. Second, the variability of exercises applied in each component, for example, for cardiovascular exercise, included those able to be performed in a seated position, which have been infrequently described in the literature. Third, other psychomotor components were incorporated. Fourth, we also prescribed PE according to the principles of FITT-VP [17], which allowed a better dosage of PE and enabled high attendance to our intervention, similar to previous reports [14]. Fifth, we can also highlight the use of valid, easily reproducible instruments often used in gerontology (CGA) and a diversity of health and wellbeing elements examined.

## 5. Conclusions

In the present investigation, a multicomponent structured PE intervention (cardiovascular, strength/power, flexibility, static and dynamic balance, other psychomotor components, and education) was prescribed according to the FITT-VP principles, at a moderate intensity, in sessions of 60 min, three times a week, and during 3 months of intervention (36 sessions in total) on a sample of Chilean older adults from the community, with controlled and compensated multimorbidity. The main results revealed significant effects on QoL, biological and biopsychosocial frailty, sarcopenia, functionality in basic, instrumental and advanced ADLs, cognitive and mood status, physical–functional condition, dynamic balance, seated and bipedal systolic blood pressure, and seated diastolic blood pressure. This intervention can serve as a pilot for RCTs or for improved PE programs or services for older adults under the auspices of existing public policy.

Consequently, developing a structured multicomponent PE intervention, according to FITT-VP principles, for older adults living in the community is important. These principals guarantee the specificity of a dose similar to a pharmacological intervention with planning over time to optimize health, thus reducing public health costs.

Likewise, in the context of gerontology, it is important to evaluate older adults with scales, questionnaires, or clinical tests such as the CGA, since these components may improve QoL and biopsychosocial health. Kinesiologists and physiotherapists are among the professionals of the interdisciplinary gerontology team with the greatest competencies for encouraging, evaluating, prescribing, educating, and managing PE among community-living older adults, whether healthy or with chronic pathologies and/or prevalent geriatric syndromes.

## Figures and Tables

**Figure 1 ijerph-19-15842-f001:**
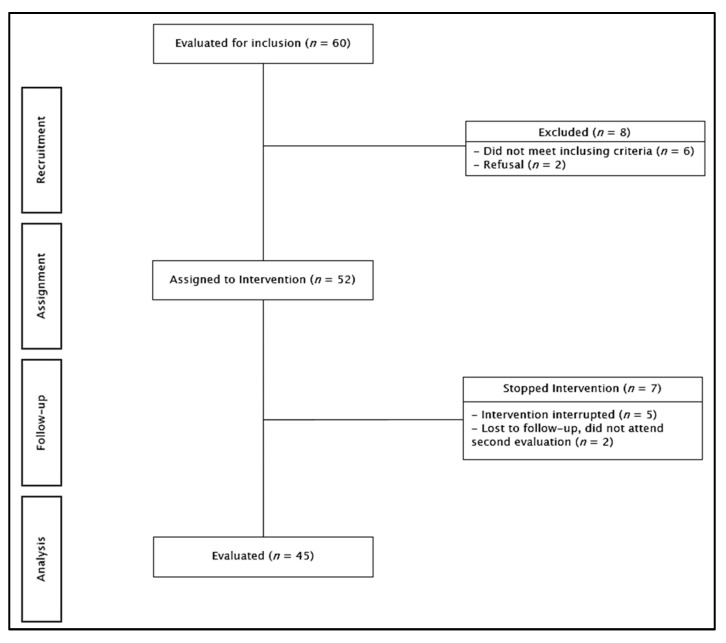
Participant flowchart.

**Figure 2 ijerph-19-15842-f002:**
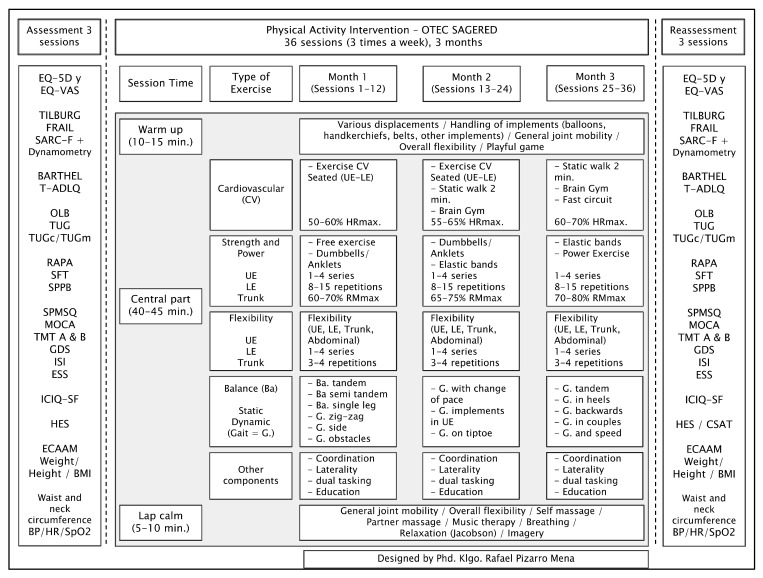
Structured multicomponent physical exercise intervention protocol in community-dwelling older adults. Abbreviations: EQ-5D: EuroQol—five dimensions; EQ-VAS: EuroQol visual analogue scale; T-ADLQ: Technology—Activities of Daily Living Questionnaire; OLB: one-leg balance; TUG: timed up and Go; TUGc: cognitive timed up and go; TUGm: manual timed up and go; RAPA: rapid assessment of physical activity; SFT: senior fitness test; SPPB: short physical performance battery; SPMSQ: short portable mental status questionnaire; MOCA: Montreal Cognitive Assessment; TMT: trail making test (parts A and B); GDS: Yesavage geriatric depression scale; ISI: insomnia severity index; ESS: Epworth Daytime Sleepiness Questionnaire; ICIQ-SF: Consultation on Incontinence Questionnaire Short-Form; HES: health empowerment scale for older adults; ECAAM: food quality survey for the older adults; BMI: body mass index; BP: blood pressure; HR: heart rate; UE: upper extremity; LE: lower extremity; CSAT: customer satisfaction survey.

**Table 1 ijerph-19-15842-t001:** Sociodemographic characteristics of intervention group participants who completed the structured multicomponent physical exercise intervention.

	Participants *n* = 45				
	*n*	Mean	SD	Min	Max
**Age**	45	70.74	7.7	60	87
**Gender**					
Female (%)	37	82.2			
Male (%)	8	17.8			
**Education**					
Primary (%)	6	13.3			
High school (%)	13	28.9			
University (%)	24	53.4			
No response	2	4.4			
**Marital Status**					
Married (%)	19	42.2			
Divorced (%)	4	8.9			
Single (%)	8	17.8			
Widow/widower (%)	12	26.7			
No response	2	4.4			
**Use of cane**					
Yes (%)	1	2.2			
No (%)	42	93.3			
No response	2	4.5			
**Number of comorbidities (list of 31)**	43	4.51	2.70	0	12
No response	2				
**Number of medications (list of 29)**	43	3.05	2.45	0	10
No response	2				
**Laterality**					
Right	42	93.33			
Left	3	6.67			
**Falls**					
Number of falls in last year	43	0.56	0.96	0	4
No response	2				

Abbreviations: *n*: sample size; SD: standard deviation; min: minimum; max: maximum.

**Table 2 ijerph-19-15842-t002:** Pre–post intervention comparison of quality of life and biopsychosocial variables among participants of a structured multicomponent physical exercise intervention.

	Participants *n* = 45				
Variables	Pre-Intervention Score	Post-Intervention Score	Change in Score	*p*-Value	Effect Size
	Mean (SD)	Mean (SD)			
Quality of life (EQ-VAS)	80.13 (21.31)	87.93 (11.21)	7.80 (3.22)	0.020 *	0.45
Biopsychosocial frailty (TILBURG)	4.36 (2.76)	3.22 (2.19)	−1.13 (2.40)	0.003 *	−0.45
Sarcopenia (SARC-F)	1.87 (1.74)	1.22 (1.35)	−0.64 (1.28)	0.002 *	−0.41
Biological frailty (FRAIL)	0.71 (0.94)	0.36 (0.57)	−0.36 (0.96)	0.017 *	−0.45
Functionality: basic activities of daily living (BARTHEL)	97.20 (0.75)	99 (0.44)	1.77 (0.83)	0.037 *	0.43
Functionality: basic, instrumental, and advanced activities of daily living (T-ADLQ total)	15.80 (10.43)	13.8 (9.80)	−2.06 (5.62)	0.021 *	−0.20
Cognitive status (SPMSQ)	8.91 (0.85)	8.98 (0.97)	0.07 (0.91)	0.627	
Cognitive status (MOCA)	23.02 (4.70)	24.41 (4.15)	1.45 (3.43)	0.007 *	0.32
Attention and psychomotor speed (TMT A)	56.97 (20.69)	56.36 (23.22)	−0.61 (20.31)	0.841	
Attention and psychomotor speed (TMT B)	154.27 (97.21)	143.73 (95.54)	−10.54 (42.90)	0.106	
Mood (GDS)	2.02 (1.84)	1.05 (1.28)	−0.98 (1.56)	0.002 *	−0.61
Sleep: insomnia (ISI)	6.84 (5.11)	6.31 (4.78)	−0.53 (4.67)	0.448	
Sleep: daytime sleepiness (ESS)	4.44 (3.10)	4.38 (3.28)	−0.07 (3.49)	0.899	
Health empowerment (HES)	38.02 (2.33)	38.64 (2.05)	0.61 (3.13)	0.199	
Food quality (ECAAM total)	77.67 (7.66)	79.02 (6.93)	1.35 (5.71)	0.118	

Abbreviations: *n*: sample size; SD: standard deviation; EQ-VAS: EuroQol visual analogue scale; T-ADLQ: Technology—Activities of Daily Living Questionnaire; SPMSQ: short portable mental status questionnaire; MOCA: Montreal Cognitive Assessment; TMT: trail making test; GDS: Yesavage geriatric depression scale; ISI: insomnia severity index; ESS: Epworth Daytime Sleepiness Questionnaire; HES: health empowerment scale for older adults; ECAAM: food quality survey for the older adults. Paired *t*-test; * *p* ˂ 0.05.

**Table 3 ijerph-19-15842-t003:** Pre–post intervention comparison for clinical variables and anthropometrics among participants of a structured multicomponent physical exercise intervention.

	Participants *n* = 45				
Variables	Pre-Intervention Score	Post-Intervention Score	Change in Score	*p*-Value	Effect Size
	Mean (SD)	Mean (SD)			
Systolic blood pressure (seated)	135.89 (17.47)	124.16 (18.63)	−11.73 (15.42)	<0.001 *	−0.65
Diastolic blood pressure (seated)	75.33 (12.47)	68.78 (10.29)	−6.56 (12.83)	0.001 *	−0.57
Systolic blood pressure (standing)	131.78 (20.07)	123.53 (17.29)	−8.24 (18.08)	0.004 *	−0.44
Diastolic blood pressure (standing)	79.22 (9.44)	77.11 (9.58)	−2.11 (8.50)	0.103	
Heart rate (seated)	73.18 (11.61)	75.56 (11.15)	2.38 (10.49)	0.136	
Heart rate (standing)	77.56 (13.09)	80.60 (12.23)	3.04 (10.25)	0.053	
Oxygen saturation (seated)	96.13 (1.95)	96.04 (1.72)	−0.09 (1.92)	0.757	
Oxygen saturation (standing)	96.53 (1.71)	96.07 (2.56)	−0.47 (2.38)	0.195	
Static balance (OLB-R (s))	24.72 (22.33)	23.75 (19.20)	−0.97 (14.54)	0.658	
Static balance (OLB-L (s))	23.18 (20.06)	25.17 (21.03)	1.99 (16.19)	0.413	
Dynamic balance (TUG (s))	10.01 (2.56)	9.42 (2.62)	−0.59 (1.97)	0.052	
Dynamic balance (TUGc: cognitive task (s))	14.27 (5.57)	12.20 (3.76)	−2.07 (4.96)	0.008 *	−0.44
Dynamic balance (TUGm: functional task (s))	13.08 (3.48)	12.21 (3.34)	−0.88 (2.92)	0.051	
**Anthropometry**					
Weight (kg)	68.41 (11.11)	67.50 (11.21)	−0.91 (2.47)	0.018 *	−0.08
Height (m)	1.56 (0.08)	1.56 (0.08)	0.003 (0.01)	0.193	
BMI	28.28 (4.23)	27.79 (4.09)	−0.49 (1.20)	0.008 *	−0.12
Waist circumference (cm)	94.08 (14.20)	95.72 (11.92)	1.64 (10.16)	0.286	
Neck circumference (cm)	36.82 (10.36)	35.23 (4.44)	−1.59 (7.94)	0.186	
**Dynamometry**					
Sarcopenia (grip strength—R)	21.58 (6.37)	21.86 (5.88)	0.28 (2.85)	0.514	
Sarcopenia (grip strength—L)	20.48 (5.93)	20.26 (5.97)	−0.23 (3.48)	0.661	

Abbreviations: SD: standard deviation; min: minimum; max: maximum; OLB: one-leg balance; R: right; L: left; TUG: timed up and go; TUGc: cognitive timed up and go; TUGm: manual timed up and go; BMI: body mass index. Paired *t*-test; * *p* ˂ 0.05.

**Table 4 ijerph-19-15842-t004:** Pre–post intervention comparison of physical–functional condition variables among participants of a structured multicomponent physical exercise intervention.

	Participants *n* = 45				
Variables	Pre-Intervention Score	Post-Intervention Score	Change in Score	*p*-Value	Effect Size
	Mean (SD)	Mean (SD)			
**SPPB**					
Side-by-side stand (s)	10.00 (0.00)	10.00 (0.00)			
Semi-tandem stand (s)	9.80 (0.90)	10.00 (0.00)	0.20 (0.90)	0.135	
Full tandem stand (s)	8.24 (3.47)	9.28 (2.13)	1.04 (3.28)	0.040 *	0.36
4 m walking speed test (s)	3.37 (0.72)	3.36 (0.84)	−0.01 (0.87)	0.913	
Sit down and stand up 5 times test (s)	14.05 (4.68)	10.76 (3.28)	−3.29 (3.43)	<0.001 *	−0.81
SPPB score (0–12)	10.02 (0.29)	11.13 (0.23)	1.11 (0.25)	<0.001 *	0.63
**SFT**					
Chair stand test (*n* repetitions)	10.58 (4.18)	12.07 (3.12)	1.49 (4.09)	0.019 *	0.40
Arm curl test (*n* repetitions)	13.51 (4.51)	15.09 (3.70)	1.58 (4.03)	0.012 *	0.38
2 min step test (*n* repetitions)	72.09 (20.05)	83.31 (20.15)	11.22 (16.91)	<0.001 *	0.55
Chair sit and reach test—preferred LE (cm)	−6.76 (8.78)	−5.51 (8.98)	1.24 (2.39)	0.001 *	0.14
Back scratch test—preferred UE (cm)	−13.83 (10.87)	−11.65 (10.92)	2.18 (5.50)	0.011 *	0.20
8 foot up and go test (s)	6.97 (2.09)	6.07 (1.90)	−0.90 (1.23)	<0.001 *	−0.45

Abbreviations: SD: standard deviation; min: minimum; max: maximum; SPPB: short physical performance battery; SFT: senior fitness test; LE: lower extremity; UE: upper extremity. Paired *t*-test; * *p* ˂ 0.05.

## Data Availability

Data are available upon request.

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
