# Peer review of "Effects of a Structured Multicomponent Physical Exercise Intervention on Quality of Life and Biopsychosocial Health among Chilean Older Adults from the Community with Controlled Multimorbidity: A Pre–Post Design"

_ijerph, 2022, doi:10.3390/ijerph192315842_

Round 1

Reviewer 1 Report

Weaknesses:

- Study design is not randomized so there might be some selection bias, I would like a little more information on the background of these individuals. Also the selection bias should be kept more in the text.

- Male/female ratio is also (due to this) not very stable

- Multitude of tests have been performed, so some of the data might be hiding in the results?

- Discussion: no discussion on the weaknesses of this study, why?

- Other than these, I think this study has been well performed and manuscript is well-written   

Reviewer 2 Report

Title

The title is appropriate but could be improved by adding brief information regarding the studied sample (i.e., apparently healthy or with comorbidities or treated chronic conditions, community-living or hospitalized, among others). It is also interesting to mention that the Chilean elderly were submitted to the intervention.

Abstract

For background, it is interesting to add a note regarding the definition of multicomponent training. I also suggest dividing this first sentence into two or three phrases (i.e., Structured multicomponent physical exercise (PE) for the elderly has shown to have benefits for physical, cognitive, social, and metabolic functioning. This type of intervention can also counteract chronic conditions and geriatric syndromes. However, little is known about the effectiveness of multicomponent PE in Chilean individuals.).

I recommend removing “evaluated with the comprehensive gerontological assessment (CGA)” from the purpose and reallocating it after the study design. In the abstract, the authors referred to the study design as quasi-experimental while in the title was presented as a pre-post design. I suggest changing this information in the title.

The methods reported in the abstract must be improved. How was evaluated the quality of life? And biopsychosocial health? It is also important to point out that the authors used several acronyms without including their definitions. I recommend including the frequency and duration of each session, among other aspects of exercise prescription. For each variable evaluated, the authors must report the tool or assessment by using parenthesis or mentioning the study protocol.

Despite being multi-component, the authors must address the modalities or types of exercises prescribed, such as aerobic and strength, stretching and aerobic, water-related activities, the recommendation for physical activity, other types of practices such as Yoga or Pilates, etc. The combination of exercises can vary and, therefore, this information must be properly reported in the abstract.

The poorly described methods lead to confusing findings. Lastly, the conclusion must be revised and improved.  

Introduction

The authors should be more specific regarding physiological and pathological aging-related changes, which can provide a better understanding of the relevance of multicomponent exercises for the reader.

I suggest not using the acronym for elderly persons. Despite of being unnecessary, it might cause confusion since the authors also used PE for physical exercise, which is very similar.

Since the authors are introducing PA and PE concepts, as well as the main benefits, and also investigating biopsychosocial health, I strongly recommend presenting data from inactivity and sedentary lifestyle in the elderly. It is also important to add the role of PA and PE for health promotion and disease prevention, which can also reduce significant costs for public health.    

Although coherent, the introduction could be improved. Some paragraphs from the discussion could be better placed in the introduction, for instance, lines 279-288.

The introduction fails in justifying the contribution of this paper to the literature. The authors cited published reviews that addressed the main effects of multicomponent PE for this population. Then, I question how this study, especially this design (quasi-experimental), can aggregate in relation to what is already known. If the main strength of this study relays on the assessment protocol, this must be addressed in the introduction. The same must be considered regarding the multicomponent exercise prescription. Therefore, the introduction needs a major revision.

Methods

For item 2.1, I recommend the authors justify this design.

In item 2.2, the authors must mention how was the recruitment. Were the evaluations mentioned related to the eligibility criteria? If the answer is no, then this information must be reallocated near the assessment protocol. What parameter was used for sample size calculation?

In figure 1, the exclusion, loss of follow-up and etc. must be placed at the right in different boxes.

In item 2.3, the authors could include whether the intervention was conducted face-to-face by individuals or in groups. The same must be done for exercise prescription, i.e., individualized based on 1RM and FEC, for instance.

Since the assessment protocol was performed before and after the intervention, I suggest adding a topic for this information prior to the description of the intervention. Thus, figure 2 will be more helpful for the reader.

Regarding figure 2, it is important adding all acronyms mentioned as footnotes.

For item 2.4, the authors must choose to maintain or remove it according to the suggestion of reporting the assessment protocol prior to the description of the intervention. It also could be interesting to divide the variables according to measured and self-reported, which introduces potential biases that need to be properly addressed in the discussion.

Results

Some of the results were duplicated instead of being interpreted, i.e., the sample was mostly elderly women, who were graduated and married. The authors should avoid citing the same results, especially using numerical data. Depending on the literature, we can be considered multimorbidity and polypharmacy, which were not properly discussed further in the manuscript.

Regarding table 2, the authors need to revise the results and provide an interpretation instead of mentioning the same data seen in the table. What health aspects changed after the intervention? For those that did not change, how can the authors explain this finding? Was expected? The same applied to the findings from tables 3 and 4.

In addition, the table must be improved. The excessive use of acronyms makes it difficult to understand the results. I suggest that the authors insert the main topics as previously made for the methods, i.e., Food Quality, Health Empowerment, etc. The assessment could appear in parentheses or using symbols with the explanation in the footnote. This can be taken into consideration for tables 3 and 4 as well, which will also allow more coherence and cohesion throughout the manuscript.   

Discussion

I recommend that the first paragraph resumes the purpose and study design, followed by the main findings and practical implications of the present study.

The discussion sounds confusing and lacks clarity. I suggest discussing the main findings and then comparing the prescription of the multicomponent exercise in the present study with the current literature.   

Paragraph lines 289-295 be reallocated near the strengths or improved by discussing the known and most prescribed multicomponent exercise interventions.  

Line 338: WhatsApp must be corrected.

The authors should address the ICF, which is known as a classification of health and health-related domains and can act as an important tool for implementing multicomponent exercise programs in community or clinical settings. Similarly, the ICF can also help to monitor the health and functioning status.

I suggest adding a topic for strengths and limitations addressed, which could be helpful for the reader. I also suggest mentioning the limitations and then the strengths. It could be clearer if the authors separate limitations and strengths into two distinct paragraphs. How do the authors counteract or overcome the limitations (during the study, when conducting the intervention, or when analyzing data)? This information also must be added if that is the case.  

Conclusion

The conclusion must be revised and coherent with the study's purpose. It is also important to make clear the sample characteristics (apparently healthy or with treated chronic conditions, community-living, Chilean, etc.) and the exercise prescription (including FITT and the types of exercises prescribed). The authors could also state future perspectives and main implications.

Reviewer 3 Report

Thank you for the opportunity to review this manuscript looking at the effect of a PE intervention on QoL and biopsychosocial health. 

On the whole, I think this is a well written and solid manuscript, however some changes are necessary before progressing this further. I provide feedback for individual sections below.

Introduction

I would consider it beneficial for you to broadly define your core concepts (Quality of Life / Biopsychosocial health) your introduction. In particular, you highlight how "biopsychosocial comprehensiveness" has not been addressed by other studies. You should define what you mean by this, and why you believe it is valuable to study this aspect.  

Method

The reporting here is detailed and the presentation of the intervention are appreciated. Again, I think the understanding and context of the CGA battery would be improved by a richer introduction as described above.

Results/Discussion

In table 2, you refer to the test used (e.g. EQ-VAS) as opposed to the construct being measured (QoL). I found this somewhat irritating, as I constantly had to flip back to the method section to remember which construct I was looking at. I suggest either labelling with only the construct, or the construct and the test used in parentheses. 

Your discussion addresses the topic of depression and deprescribing. However, I feel these topics emerge a bit 'out of nowhere'. It would be helpful if you addressed/contextualised these issues (briefly) in the introduction as well. Indeed, in general, you might want to consider providing an additional background section on the biopsychosocial health and related issues, as many of the themes in the discussion could benefit from additional preliminary context. 
